# Learning Stationary Time Series using Gaussian Processes with Nonparametric Kernels

**Felipe Tobar**
ftobar@dim.uchile.cl
Center for Mathematical Modeling
Universidad de Chile

**Thang D. Bui**
tdb40@cam.ac.uk
Department of Engineering
University of Cambridge

**Richard E. Turner**
ret26@cam.ac.uk
Department of Engineering
University of Cambridge

## Abstract

We introduce the Gaussian Process Convolution Model (GPCM), a two-stage non-parametric generative procedure to model stationary signals as the convolution between a continuous-time white-noise process and a continuous-time linear filter drawn from Gaussian process. The GPCM is a continuous-time nonparametric-window moving average process and, conditionally, is itself a Gaussian process with a nonparametric kernel defined in a probabilistic fashion. The generative model can be equivalently considered in the frequency domain, where the power spectral density of the signal is specified using a Gaussian process. One of the main contributions of the paper is to develop a novel variational free-energy approach based on inter-domain inducing variables that efficiently learns the continuous-time linear filter and infers the driving white-noise process. In turn, this scheme provides closed-form probabilistic estimates of the covariance kernel and the noise-free signal both in denoising and prediction scenarios. Additionally, the variational inference procedure provides closed-form expressions for the approximate posterior of the spectral density given the observed data, leading to new Bayesian nonparametric approaches to spectrum estimation. The proposed GPCM is validated using synthetic and real-world signals.

## 1 Introduction

Gaussian process (GP) regression models have become a standard tool in Bayesian signal estimation due to their expressiveness, robustness to overfitting and tractability [1]. GP regression begins with a prior distribution over functions that encapsulates *a priori* assumptions, such as smoothness, stationarity or periodicity. The prior is then updated by incorporating information from observed data points via their likelihood functions. The result is a posterior distribution over functions that can be used for prediction. Critically for this work, the posterior and therefore the resultant predictions, is sensitive to the choice of prior distribution. The form of the prior covariance function (or kernel) of the GP is arguably the central modelling choice. Employing a simple form of covariance will limit the GP's capacity to generalise. The ubiquitous radial basis function or squared exponential kernel, for example, implies prediction is just a local smoothing operation [2, 3]. Expressive kernels are needed [4, 5], but although kernel design is widely acknowledged as pivotal, it typically proceeds via a "black art" in which a particular functional form is hand-crafted using intuitions about the application domain to build a kernel using simpler primitive kernels as building blocks (e.g. [6]).

Recently, some sophisticated automated approaches to kernel design have been developed that construct kernel mixtures on the basis of incorporating different measures of similarity [7, 8], or more generally by both adding and multiplying kernels, thus mimicking the way in which a human would search for the best kernel [5]. Alternatively, a flexible parametric kernel can be used as in the case of the spectral mixture kernels, where the power spectral density (PSD) of the GP is parametrised by a mixture of Gaussians [4].

We see two problems with this general approach: The first is that computational tractability limits the complexity of the kernels that can be designed in this way. Such constraints are problematic when searching over kernel combinations and to a lesser extent when fitting potentially large numbers of kernel hyperparameters. Indeed, many naturally occurring signals contain more complex structure than can comfortably be entertained using current methods, time series with complex spectra like sounds being a case in point [9, 10]. The second limitation is that hyperparameters of the kernel are typically fit by maximisation of the model marginal likelihood. For complex kernels with large numbers of hyperparameters, this can easily result in overfitting rearing its ugly head once more (see sec. 4.2).

This paper attempts to remedy the existing limitations of GPs in the time series setting using the same rationale by which GPs were originally developed. That is, *kernels* themselves are treated nonparametrically to enable flexible forms whose complexity can grow as more structure is revealed in the data. Moreover, approximate Bayesian inference is used for estimation, thus side-stepping problems with model structure search and protecting against overfitting. These benefits are achieved by modelling time series as the output of a linear and time-invariant system defined by a convolution between a white-noise process and a continuous-time linear filter. By considering the filter to be drawn from a GP, the expected second-order statistics (and, as a consequence, the spectral density) of the output signal are defined in a nonparametric fashion. The next section presents the proposed model, its relationship to GPs and how to sample from it. In Section 3 we develop an analytic approximate inference method using state-of-the-art variational free-energy approximations for performing inference and learning. Section 4 shows simulations using both synthetic and real-world datasets. Finally, Section 5 presents a discussion of our findings.

## 2  Regression model: Convolving a linear filter and a white-noise process

We introduce the Gaussian Process Convolution Model (GPCM) which can be viewed as constructing a distribution over functions $f(t)$ using a two-stage generative model. In the first stage, a continuous filter function $h(t) : \mathbb{R} \mapsto \mathbb{R}$ is drawn from a GP with covariance function $\mathcal{K}_h(t_1, t_2)$. In the second stage, the function $f(t)$ is produced by convolving the filter with continuous time white-noise $x(t)$. The white-noise can be treated informally as a draw from a GP with a delta-function covariance,[1]

$$h(t) \sim \mathcal{G}P(\mathbf{0}, \mathcal{K}_h(t_1, t_2)), \quad x(t) \sim \mathcal{G}P(\mathbf{0}, \sigma_x^2 \delta(t_1 - t_2)), \quad f(t) = \int_{\mathbb{R}} h(t - \tau) x(\tau) \mathrm{d}\tau. \quad (1)$$

This family of models can be motivated from several different perspectives due to the ubiquity of continuous-time linear systems.

First, the model relates to linear time-invariant (LTI) systems [12]. The process $x(t)$ is the input to the LTI system, the function $h(t)$ is the system's impulse response (which is modelled as a draw from a GP) and $f(t)$ is its output. In this setting, as an LTI system is entirely characterised by its impulse response [12], model design boils down to identifying a suitable function $h(t)$. A second perspective views the model through the lens of differential equations, in which case $h(t)$ can be considered to be the Green's function of a system defined by a linear differential equation that is driven by white-noise. In this way, the prior over $h(t)$ implicitly defines a prior over the coefficients of linear differential equations of potentially infinite order [13]. Third, the GPCM can be thought of as a continuous-time generalisation of the discrete-time moving average process in which the window is potentially infinite in extent and is produced by a GP prior [14].

A fourth perspective relates the GPCM to standard GP models. Consider the filter $h(t)$ to be known. In this case the process $f(t)|h$ is distributed according to a GP, since $f(t)$ is a linear combination of Gaussian random variables. The mean function $m_{f|h}(f(t))$ and covariance function $\mathcal{K}_{f|h}(t_1, t_2)$ of the random variable $f|h, t \in \mathbb{R}$, are then stationary and given by $m_{f|h}(f(t)) = \mathbb{E}\left[f(t)|h\right] = \int_{\mathbb{R}} h(t - \tau)\mathbb{E}\left[x(\tau)\right] \mathrm{d}\tau = 0$ and

$$\mathcal{K}_{f|h}(t_1, t_2) = \mathcal{K}_{f|h}(t) = \int_{\mathbb{R}} h(s)h(s + t)\mathrm{d}s = (h(t) * h(-t))(t) \quad (2)$$

that is, the convolution between the filter $h(t)$ and its mirrored version with respect to $t = 0$ — see sec. 1 of the supplementary material for the full derivation.

Since $h(t)$ is itself is drawn from a nonparametric prior, the presented model (through the relationship above) induces a prior over nonparametric kernels. A particular case is obtained when $h(t)$ is chosen as the basis expansion of a reproducing kernel Hilbert space [15] with parametric kernel (e.g., the squared exponential kernel), whereby $\mathcal{K}_{f|h}$ becomes such a kernel.

A fifth perspective considers the model in the frequency domain rather than the time domain. Here the continuous-time linear filter shapes the spectral content of the input process $x(t)$. As $x(t)$ is white-noise, it has positive PSD at all frequencies, which can potentially influence $f(t)$. More precisely, since the PSD of $f|h$ is given by the Fourier transform of the covariance function (by the Wiener–Khinchin theorem [12]), the model places a nonparametric prior over the PSD, given by $\mathcal{F}(\mathcal{K}_{f|h}(t))(\omega) = \int_{\mathbb{R}} \mathcal{K}_{f|h}(t)e^{-j\omega t}\mathrm{d}t = |\tilde{h}(\omega)|^2$, where $\tilde{h}(\omega) = \int_{\mathbb{R}} h(t)e^{-j\omega t}\mathrm{d}t$ is the Fourier transform of the filter.

Armed with these different theoretical perspectives on the GPCM generative model, we next focus on how to design appropriate covariance functions for the filter.

## 2.1 Sensible and tractable priors over the filter function

Real-world signals have finite power (which relates to the stability of the system) and potentially complex spectral content. How can such knowledge be built into the filter covariance function $\mathcal{K}_h(t_1, t_2)$? To fulfil these conditions, we model the linear filter $h(t)$ as a draw from a squared exponential GP that is multiplied by a Gaussian window (centred on zero) in order to restrict its extent. The resulting *decaying squared exponential* (DSE) covariance function is given by a squared exponential (SE) covariance pre- and post-multiplied by $e^{-\alpha t_1^2}$ and $e^{-\alpha t_2^2}$ respectively, that is,

$$\mathcal{K}_h(t_1, t_2) = K_{\mathrm{DSE}}(t_1, t_2) = \sigma_h^2 e^{-\alpha t_1^2} e^{-\gamma(t_1 - t_2)^2} e^{-\alpha t_2^2}, \ \alpha, \gamma, \sigma_h > 0. \tag{3}$$

With the GP priors for $x(t)$ and $h(t)$, $f(t)$ is zero-mean, stationary and has a variance $\mathbb{E}[f^2(t)] = \sigma_x^2 \sigma_h^2 \sqrt{\pi/(2\alpha)}$. Consequently, by Chebyshev's inequality, $f(t)$ is stochastically bounded, that is, $\Pr(|f(t)| \geq T) \leq \sigma_x^2 \sigma_h^2 \sqrt{\pi/(2\alpha)}T^{-2}$, $T \in \mathbb{R}$. Hence, the exponential decay of $K_{\mathrm{DSE}}$ (controlled by $\alpha$) plays a key role in the finiteness of the integral in eq. (1) — and, consequently, of $f(t)$.

Additionally, the DSE model for the filter $h(t)$ provides a flexible prior distribution over linear systems, where the hyperparameters have physical meaning: $\sigma_h^2$ controls the power of the output $f(t)$; $1/\sqrt{\gamma}$ is the characteristic timescale over which the filter varies that, in turn, determines the typical frequency content of the system; finally, $1/\sqrt{\alpha}$ is the temporal extent of the filter which controls the length of time correlations in the output signal and, equivalently, the bandwidth characteristics in the frequency domain.

Although the covariance function is flexible, its Gaussian form facilitates analytic computation that will be leveraged when (approximately) sampling from the DSE-GPCM and performing inference. In principle, it is also possible in the framework that follows to add causal structure into the covariance function so that only causal filters receive non-zero prior probability density, but we leave that extension for future work.

## 2.2 Sampling from the model

Exact sampling from the proposed model in eq. (1) is not possible, since it requires computation of the convolution between infinite dimensional processes $h(t)$ and $x(t)$. It is possible to make some analytic progress by considering, instead, the GP formulation of the GPCM in eq. (2) and noting that sampling $f(t)|h \sim \mathcal{GP}(\mathbf{0}, \mathcal{K}_{f|h})$ only requires knowledge of $\mathcal{K}_{f|h} = h(t) * h(-t)$ and therefore avoids explicit representation of the troublesome white-noise process $x(t)$. Further progress requires approximation. The first key insight is that $h(t)$ can be sampled at a finite number of locations $\mathbf{h} = h(\mathbf{t}) = [h(t_1), \ldots, h(t_{N_\mathbf{h}})]$ using a multivariate Gaussian and then exact analytic inference can be performed to infer the entire function $h(t)$ (via noiseless GP regression). Moreover, since the filter is drawn from the DSE kernel $h(t) \sim \mathcal{GP}(\mathbf{0}, K_{\mathrm{DSE}})$ it is, with high probability, temporally limited in extent and smoothly varying. Therefore, a relatively small number of samples $N_\mathbf{h}$ can potentially enable accurate estimates of $h(t)$. The second key insight is that it is possible,

when using the DSE kernel, to analytically compute the expected value of the covariance of $f(t)|\mathbf{h}$, $\mathcal{K}_{f|\mathbf{h}} = \mathbb{E}[\mathcal{K}_{f|h}|\mathbf{h}] = \mathbb{E}[h(t) * h(-t)|\mathbf{h}]$ as well as the uncertainty in this quantity. The more values the latent process $h$ we consider, the lower the uncertainty in $h$ and, as a consequence, $\mathcal{K}_{f|\mathbf{h}} \to \mathcal{K}_{f|h}$ almost surely. This is an example of a Bayesian numerical integration method since the approach maintains knowledge of its own inaccuracy [16].

In more detail, the kernel approximation $\mathcal{K}_{f|\mathbf{h}}(t_1, t_2)$ is given by:

$$\mathbb{E}[\mathcal{K}_{f|h}(t_1,t_2)|\mathbf{h}] = \mathbb{E}\left[\int_{\mathbb{R}} h(t_1-\tau)h(t_2-\tau)\mathrm{d}\tau\middle|\mathbf{h}\right] = \int_{\mathbb{R}} \mathbb{E}\left[h(t_1-\tau)h(t_2-\tau)|\mathbf{h}\right]\mathrm{d}\tau$$

$$= \int_{\mathbb{R}} K_{\mathrm{DSE}}(t_1-\tau, t_2-\tau)\mathrm{d}\tau + \sum_{r,s=1}^{N_g} M_{r,s}\int_{\mathbb{R}} K_{\mathrm{DSE}}(t_1-\tau, \mathbf{t}_r)K_{\mathrm{DSE}}(\mathbf{t}_s, t_2-\tau)\mathrm{d}\tau$$

where $M_{r,s}$ is the $(r,s)^{\mathrm{th}}$ entry of the matrix $(\mathbf{K}^{-1}\mathbf{h}\mathbf{h}^T\mathbf{K}^{-1} - \mathbf{K}^{-1})$, $\mathbf{K} = K_{\mathrm{DSE}}(\mathbf{t}, \mathbf{t})$. The kernel approximation and its Fourier transform, i.e., the PSD, can be calculated in closed form (see sec. 2 in the supplementary material). Fig. 1 illustrates the generative process of the proposed model.

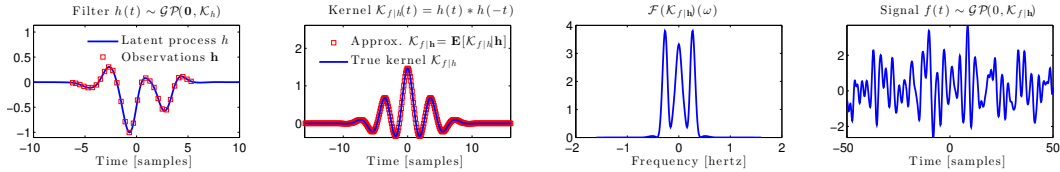

Figure 1: Sampling from the proposed regression model. From left to right: filter, kernel, power spectral density and sample of the output $f(\cdot)$.

## 3 Inference and learning using variational methods

One of the main contributions of this paper is to devise a computationally tractable method for learning the filter $h(t)$ (known as system identification in the control community [17]) and inferring the white-noise process $x(t)$ from a noisy dataset $\mathbf{y} \in \mathbb{R}^N$ produced by their convolution and additive Gaussian noise, $y(t) = f(t) + \epsilon(t) = \int_{\mathbb{R}} h(t-\tau)x(\tau)\mathrm{d}\tau + \epsilon(t)$, $\epsilon(t) \sim \mathcal{N}(0,\sigma_\epsilon^2)$. Performing inference and learning is challenging for three reasons: First, the convolution means that *each* observed datapoint depends on the *entire* unknown filter and white-noise process, which are infinite-dimensional functions. Second, the model is non-linear in the unknown functions since the filter and the white-noise multiply one another in the convolution. Third, continuous-time white-noise must be handled with care since formally it is only well-behaved inside integrals.

We propose a variational approach that addresses these three problems. First, the convolution is made tractable by using variational inducing variables that summarise the infinite dimensional latent functions into finite dimensional inducing points. This is the same approach that is used for scaling GP regression [18]. Second, the product non-linearity is made tractable by using a structured mean-field approximation and leveraging the fact that the posterior is conditionally a GP when $x(t)$ or $h(t)$ is fixed. Third, the direct representation of white-noise process is avoided by considering a set of inducing variables instead, which are related to $x(t)$ via an integral transformation (so-called inter-domain inducing variables [19]). We outline the approach below.

In order to form the variational inter-domain approximation, we first expand the model with additional variables. We use $X$ to denote the set of all integral transformations of $x(t)$ with members $u_x(t) = \int w(t,\tau)x(\tau)\mathrm{d}\tau$ (which includes the original white-noise process when $w(t,\tau) = \delta(t-\tau)$) and identically define the set $H$ with members $u_h(t) = \int w(t,\tau)h(\tau)\mathrm{d}\tau$. The variational lower bound of the model evidence can be applied to this augmented model[2] using Jensen's inequality

$$\mathcal{L} = \log p(\mathbf{y}) = \log\int p(\mathbf{y}, H, X)\mathrm{d}H\mathrm{d}X \geq \int q(H,X)\log\frac{p(\mathbf{y},H,X)}{q(H,X)}\mathrm{d}H\mathrm{d}X = \mathcal{F} \quad (4)$$

here $q(H, X)$ is any variational distribution over the sets of processes $X$ and $H$. The bound can be written as the difference between the model evidence and the KL divergence between the variational distribution over all integral transformed processes and the true posterior, $\mathcal{F} = \mathcal{L} - \text{KL}[q(H, X)||p(X, H|\mathbf{y})]$. The bound is therefore saturated when $q(H, X) = p(X, H|\mathbf{y})$, but this is intractable. Instead, we choose a simpler parameterised form, similar in spirit to that used in the approximate sampling procedure, that allows us to side-step these difficulties. In order to construct the variational distribution, we first partition the set $X$ into the original white-noise process, a finite set of variables called inter-domain inducing points $\boldsymbol{u}_x$ that will be used to parameterise the approximation and the remaining variables $X_{\neq x, \boldsymbol{u}_x}$, so that $X = \{x, \boldsymbol{u}_x, X_{\neq x, \boldsymbol{u}_x}\}$. The set $H$ is partitioned identically $H = \{h, \boldsymbol{u}_h, H_{\neq h, \boldsymbol{u}_h}\}$. We then choose a variational distribution $q(H, X)$ that mirrors the form of the joint distribution,

$$p(\mathbf{y}, H, X) = p(x, X_{\neq x, \boldsymbol{u}_x}|\boldsymbol{u_x})p(h, H_{\neq h, \boldsymbol{u}_h}|\boldsymbol{u_h})p(\boldsymbol{u}_x)p(\boldsymbol{u}_h)p(\mathbf{y}|h, x)$$
$$q(H, X) = p(x, X_{\neq x, \boldsymbol{u}_x}|\boldsymbol{u}_x)p(h, H_{\neq h, \boldsymbol{u}_h}|\boldsymbol{u_h})q(\boldsymbol{u}_x)q(\boldsymbol{u}_h) = q(H)q(X).$$

This is a structured mean-field approximation [21]. The approximating distribution over the inducing points $q(\boldsymbol{u}_x)q(\boldsymbol{u}_h)$ is chosen to be a multivariate Gaussian (the optimal parametric form given the assumed factorisation). Intuitively, the variational approximation implicitly constructs a surrogate GP regression problem, whose posterior $q(\boldsymbol{u}_x)q(\boldsymbol{u}_h)$ induces a predictive distribution that best captures the true posterior distribution as measured by the KL divergence.

Critically, the resulting bound is now tractable as we will now show. First, note that the shared prior terms in the joint and approximation cancel leading to an elegant form,

$$\mathcal{F} = \int q(h, x, \boldsymbol{u}_h, \boldsymbol{u}_x) \log \frac{p(\mathbf{y}|h, x)p(\boldsymbol{u}_h)p(\boldsymbol{u}_x)}{q(\boldsymbol{u}_h)q(\boldsymbol{u}_x)} \mathrm{d}h\mathrm{d}x\mathrm{d}\boldsymbol{u}_h\mathrm{d}\boldsymbol{u}_x \tag{5}$$
$$= \mathbb{E}_q\left[\log p(\mathbf{y}|h, x)\right] - \text{KL}[q(\boldsymbol{u}_h)||p(\boldsymbol{u}_h)] - \text{KL}[q(\boldsymbol{u}_x)||p(\boldsymbol{u}_x)]. \tag{6}$$

The last two terms in the bound are simple to compute being KL divergences between multivariate Gaussians. The first term, the average of the log-likelihood terms with respect to the variational distribution, is more complex,

$$\mathbb{E}_q\left[\log p(\mathbf{y}|h, x)\right] = -\frac{N}{2} \log(2\pi\sigma_\epsilon^2) - \frac{1}{2\sigma_\epsilon^2} \sum_{i=1}^N \mathbb{E}_q\left[\left(y(t_i) - \int_{\mathbb{R}} h(t_i - \tau)x(\tau)\mathrm{d}\tau\right)^2\right].$$

Computation of the variational bound therefore requires the first and second moments of the convolution under the variational approximation. However, these can be computed analytically for particular choices of covariance function such as the DSE, by taking the expectations inside the integral (this is analogous to variational inference for the Gaussian Process Latent Variable Model [22]). For example, the first moment of the convolution is

$$\mathbb{E}_q\left[\int_{\mathbb{R}} h(t_i - \tau)x(\tau)\mathrm{d}\tau\right] = \int_{\mathbb{R}} \mathbb{E}_{q(h, \boldsymbol{u}_h)}\left[h(t_i - \tau)\right]\mathbb{E}_{q(x, \boldsymbol{u}_x)}[x(\tau)]\mathrm{d}\tau \tag{7}$$

where the expectations take the form of the predictive mean in GP regression, $\mathbb{E}_{q(h, \boldsymbol{u}_h)}\left[h(t_i - \tau)\right] = K_{h, \boldsymbol{u}_h}(t_i - \tau)K_{\boldsymbol{u}_h, \boldsymbol{u}_h}^{-1}\mu_{\boldsymbol{u}_h}$ and $\mathbb{E}_{q(x, \boldsymbol{u}_x)}[x(\tau)] = K_{x, \boldsymbol{u}_x}(\tau)K_{\boldsymbol{u}_x, \boldsymbol{u}_x}^{-1}\mu_{\boldsymbol{u}_x}$ where $\{K_{h, \boldsymbol{u}_h}, K_{\boldsymbol{u}_h, \boldsymbol{u}_h}, K_{x, \boldsymbol{u}_x}, K_{\boldsymbol{u}_x, \boldsymbol{u}_x}\}$ are the covariance functions and $\{\mu_{\boldsymbol{u}_h}, \mu_{\boldsymbol{u}_x}\}$ are the means of the approximate variational posterior. Crucially, the integral is tractable if the covariance functions can be convolved analytically, $\int_{\mathbb{R}} K_{h, \boldsymbol{u}_h}(t_i - \tau)K_{x, \boldsymbol{u}_x}(\tau)\mathrm{d}\tau$, which is the case for the SE and DSE covariances - see sec. 4 of the supplementary material for the derivation of the variational lower bound.

The fact that it is possible to compute the first and second moments of the convolution under the approximate posterior means that it is also tractable to compute the mean of the posterior distribution over the kernel, $\mathbb{E}_q\left[\mathcal{K}_{f|h}(t_1, t_2)\right] = \mathbb{E}_q\left[\int_{\mathbb{R}} h(t_1 - \tau)h(t_2 - \tau)\mathrm{d}\tau\right]$ and the associated error-bars. The method therefore supports full probabilistic inference and learning for nonparametric kernels, in addition to extrapolation, interpolation and denoising in a tractable manner. The next section discusses sensible choices for the integral transforms that define the inducing variables $\boldsymbol{u}_h$ and $\boldsymbol{u}_x$.

### 3.1   Choice of the inducing variables $u_h$ and $u_x$

In order to choose the domain of the inducing variables, it is useful to consider inference for the white-noise process given a fixed window $h(t)$. Typically, we assume that the window $h(t)$ is

smoothly varying, in which case the data $y(t)$ are only determined by the low-frequency content of the white-noise; conversely in inference, the data can only reveal the low frequencies in $x(t)$. In fact, since a continuous time white-noise process contains power at all frequencies and infinite power in total, most of the white-noise content will be undeterminable, as it is suppressed by the filter (or *filtered out*). However, for the same reason, these components do not affect prediction of $f(t)$.

Since we can only learn the low-frequency content of the white-noise and this is all that is important for making predictions, we consider inter-domain inducing points formed by a Gaussian integral transform, $u_x = \int_{\mathbb{R}} \exp\left(-\frac{1}{2l^2}(t_x - \tau)^2\right) x(\tau) \mathrm{d}\tau$. These inducing variables represent a local estimate of the white-noise process $x$ around the inducing location $t_x$ considering a Gaussian window, and have a squared exponential covariance by construction (these covariances are shown in sec. 3 of the supplementary material). In spectral terms, the process $u_x$ is a low-pass version of the true process $x$. The variational parameters $l$ and $t_x$ affect the approximate posterior and can be optimised using the free-energy, although this was not investigated here to minimise computational overhead. For the inducing variables $u_h$ we chose not to use the flexibility of the inter-domain parameterisation and, instead, place the points in the same domain as the window.

## 4 Experiments

The DSE-GPCM was tested using synthetic data with known statistical properties and real-world signals. The aim of these experiments was to validate the new approach to learn covariance functions and PSDs while also providing error bars for the estimates, and to compare it against alternative parametric and nonparametric approaches.

### 4.1 Learning known parametric kernels

We considered Gaussian processes with standard, parametric covariance kernels and verified that our method is able to infer such kernels. Gaussian processes with squared exponential (GP-SE) and spectral mixture (GP-SM) kernels, both of unit variance, were used to generate two time series on the region [-44, 44] uniformly sampled at 10 Hz (i.e., 880 samples). We then constructed the observation signal by adding unit-variance white-noise. The experiment then consisted of (i) learning the underlying kernel, (ii) estimating the latent process and (iii) performing imputation by removing observations in the region [-4.4, 4.4] (10% of the observations).

Fig. 2 shows the results for the GP-SE case. We chose 88 inducing points for $u_x$, that is, 1/10 of the samples to be recovered and 30 for $u_h$; the hyperparameters in eq. (2) were set to $\gamma = 0.45$ and $\alpha = 0.1$, so as to allow for an uninformative prior on $h(t)$. The variational objective $\mathcal{F}$ was optimised with respect to the hyperparameter $\sigma_h$ and the variational parameters $\mu_h, \mu_x$ (means) and the Cholesky factors of $C_h, C_x$ (covariances) using conjugate gradients. The true SE kernel was reconstructed from the noisy data with an accuracy of 5%, while the estimation mean squared error (MSE) was within 1% of the (unit) noise variance for both the true GP-SE and the proposed model.

Fig. 3 shows the results for the GP-SM time series. Along the lines of the GP-SE case, the reconstruction of the true kernel and spectrum is remarkably accurate and the estimate of the latent process has virtually the same mean square error (MSE) as the true GP-SM model. These toy results indicate that the variational inference procedure can work well, in spite of known biases [23].

### 4.2 Learning the spectrum of real-world signals

The ability of the DSE-GPCM to provide Bayesian estimates of the PSD of real-world signals was verified next. This was achieved through a comparison of the proposed model to (i) the spectral mixture kernel (GP-SM) [4], (ii) tracking the Fourier coefficients using a Kalman filter (Kalman-Fourier [24]), (iii) the Yule-Walker method and (iv) the periodogram [25].

We first analysed the Mauna Loa monthly $CO_2$ concentration (de-trended). We considered the GP-SM with 4 and 10 components, Kalman-Fourier with a partition of 500 points between zero and the Nyquist frequency, Yule-Walker with 250 lags and the raw periodogram. All methods used all the data and each PSD estimate was normalised w.r.t its maximum (shown in fig. 4). All methods identified the three main frequency peaks at [0, year$^{-1}$, 2year$^{-1}$]; however, notice that the Kalman-Fourier method does not provide sharp peaks and that GP-SM places Gaussians on frequencies with

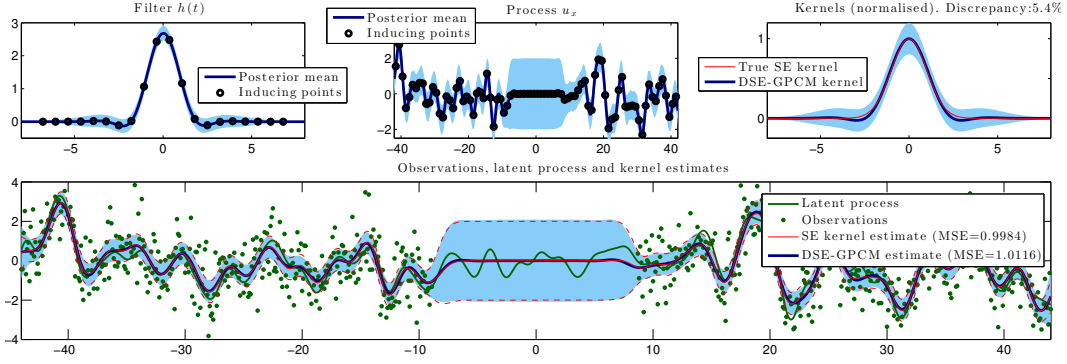

Figure 2: Joint learning of an SE kernel and data imputation using the proposed DSE-GPCM approach. **Top:** filter $h(t)$ and inducing points $u_h$ (left), filtered white-noise process $u_x$ (centre) and learnt kernel (right). **Bottom:** Latent signal and its estimates using both the DSE-GPCM and the true model (GP-SE). Confidence intervals are shown in light blue (DSE-GPCM) and in between dashed red lines (GP-SE) and they correspond to 99.7% for the kernel and 95% otherwise.

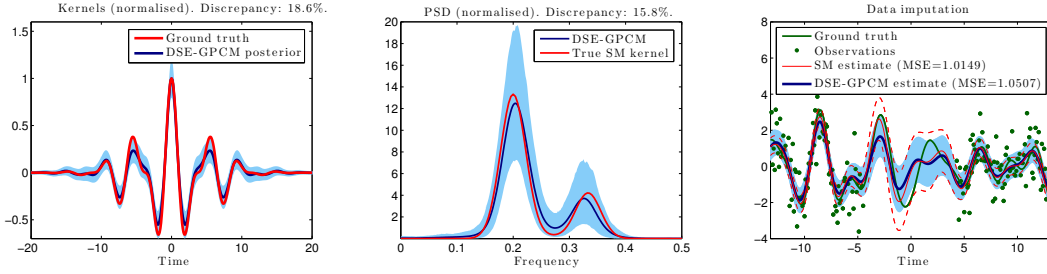

Figure 3: Joint learning of an SM kernel and data imputation using a nonparametric kernel. True and learnt kernel (left), true and learnt spectra (centre) and data imputation region (right).

negligible power — this is a known drawback of the GP-SM approach: it is sensitive to initialisation and gets trapped in noisy frequency peaks (in this experiment, the centres of the GP-SM were initialised as multiples of one tenth of the Nyquist frequency). This example shows that the GP-SM can overfit noise in training data. Conversely, observe how the proposed DSE-GPCM approach (with $N_{\mathbf{h}} = 300$ and $N_{\mathbf{x}} = 150$) not only captured the first three peaks but also the *spectral floor* and placed meaningful error bars (90%) where the raw periodogram laid.

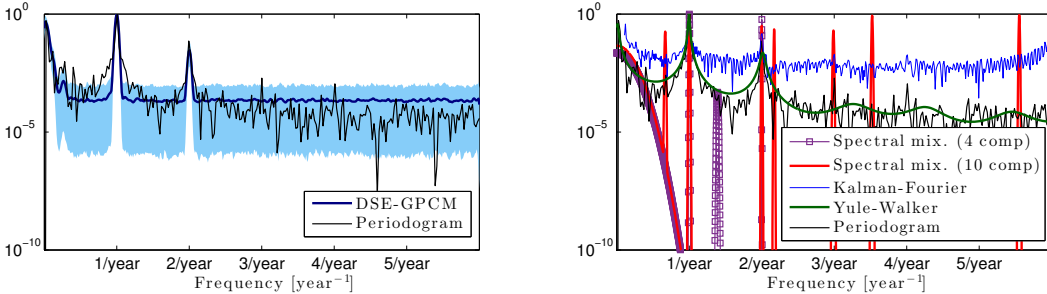

Figure 4: Spectral estimation of the Mauna Loa $CO_2$ concentration. DSE-GPCM with error bars (90%) is shown with the periodogram at the left and all other methods at the right for clarity.

The next experiment consisted of recovering the spectrum of an audio signal from the TIMIT corpus, composed of 1750 samples (at 16kHz), only using an irregularly-sampled 20% of the available data. We compared the proposed DSE-GPCM method to GP-SM (again 4 and 10 components) and Kalman-Fourier; we used the periodogram and the Yule-Walker method as benchmarks, since these

methods cannot handle unevenly-sampled data (therefore, they used all the data). Besides the PSD, we also computed the learnt kernel, shown alongside the autocorrelation function in fig. 5 (left).

Due to its sensitivity to initial conditions, the centres of the GP-SM were initialised every 100Hz (the harmonics of the signal are approximately every 114Hz); however, it was only with 10 components that the GP-SM was able to find the four main lobes of the PSD. Notice also how the DSE-GPCM accurately finds the main lobes, both in location and width, together with the 90% error bars.

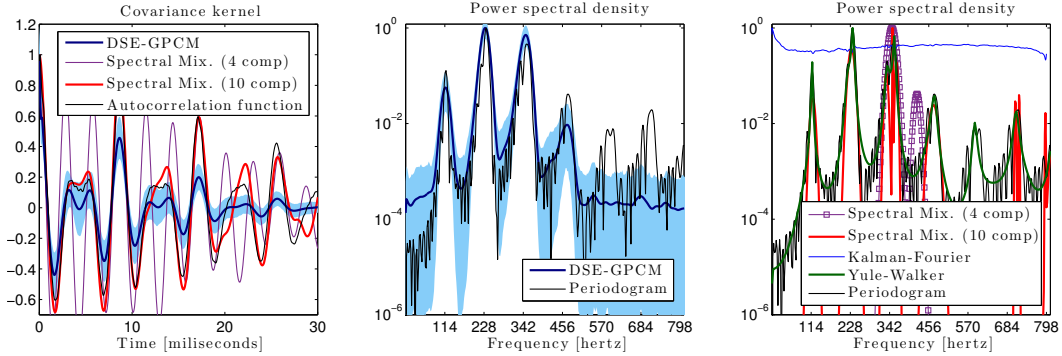

Figure 5: Audio signal from TIMIT. Induced kernel of DSE-GPCM and GP-SM alongside autocorrelation function (left). PSD estimate using DSE-GPCM and raw periodogram (centre). PSD estimate using GP-SM, Kalman-Fourier, Yule-Walker and raw periodogram (right).

## 5   Discussion

The Gaussian Process Convolution Model (GPCM) has been proposed as a generative model for stationary time series based on the convolution between a filter function and a white-noise process. Learning the model from data is achieved via a novel variational free-energy approximation, which in turn allows us to perform predictions and inference on both the covariance kernel and the spectrum in a probabilistic, analytically and computationally tractable manner. The GPCM approach was validated in the recovery of spectral density from non-uniformly sampled time series; to our knowledge, this is the first probabilistic approach that places nonparametric prior over the spectral density itself and which recovers a posterior distribution over that density directly from the time series.

The encouraging results for both synthetic and real-world data shown in sec. 4 serve as a proof of concept for the nonparametric design of covariance kernels and PSDs using convolution processes. In this regard, extensions of the presented model can be identified in the following directions: First, for the proposed GPCM to have a desired performance, the number of inducing points $\mathbf{u}_h$ and $\mathbf{u}_x$ needs to be increased with the (i) high frequency content and (ii) range of correlations of the data; therefore, to avoid the computational overhead associated to large quantities of inducing points, the filter prior or the inter-domain transformation can be designed to have a specific harmonic structure and therefore focus on a target spectrum. Second, the algorithm can be adapted to handle longer time series, for instance, through the use of tree-structured approximations [26]. Third, the method can also be extended beyond time series to operate on higher-dimensional input spaces; this can be achieved by means of a factorisation of the latent kernel, whereby the number of inducing points for the filter only increases linearly with the dimension, rather than exponentially.

#### Acknowledgements

Part of this work was carried out when F.T. was with the University of Cambridge. F.T. thanks CONICYT-PAI grant 82140061 and Basal-CONICYT Center for Mathematical Modeling (CMM). R.T. thanks EPSRC grants EP/L000776/1 and EP/M026957/1. T.B. thanks Google. We thank Mark Rowland, Shane Gu and the anonymous reviewers for insightful feedback.

## Footnotes

[1] Here we use informal notation common in the GP literature. A more formal treatment would use stochastic integral notation [11], which replaces the differential element $x(\tau)\mathrm{d}\tau = \mathrm{d}W(\tau)$, so that eq. (1) becomes a stochastic integral equation (w.r.t. the Brownian motion $W$).

[2]This formulation can be made technically rigorous for latent *functions* [20], but we do not elaborate on that here to simplify the exposition.

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
