[Supplementary Material]



# Learning Stationary Time Series using Gaussian Processes with Nonparametric Kernels [1]

Felipe Tobar, Thang D. Bui and Richard E. Turner

The following identities will be used in the derivation in this document

$$\int_{\mathbb{R}} \exp\left(-a_0\tau^2 + b_0\tau + c_0\right) \mathrm{d}\tau = \sqrt{\tfrac{\pi}{a_0}} \exp\left(c_0 + \frac{b_0^2}{4a_0}\right) \tag{1}$$

$$\int_{\mathbb{R}} \exp\left(-a_1\tau_1^2 - a_2\tau_2^2 + b_{12}\tau_1\tau_2 + b_1\tau_1 + b_2\tau_2 + c_1\right)\mathrm{d}\tau = \frac{2\pi}{\sqrt{4a_1a_2 - b_{12}^2}}\exp\left(c_1 + \frac{a_1b_1^2 + a_2b_2^2 + b_1b_{12}b_2}{4a_1a_2 - b_{12}^2}\right) \tag{2}$$

where $a_0, a_1, a_2 > 0$ and $b_0, b_1, b_2, b_{12}, c_0, c_1 \in \mathbb{R}$ are constants.

## 1 Derivation of the kernel $\mathcal{K}_{f|h}$ — given the filter function $h$

The covariance function $\mathcal{K}_{f|h}(t_1, t_2)$ of the random variable $f|h, t \in \mathbb{R}$ is given by

$$\mathcal{K}_{f|h}(t_1, t_2) = \mathbb{E}\left[f(t_1)f(t_2)|h\right]$$

$$= \mathbb{E}\left[\left.\int_{\mathbb{R}^2} h(t_1 - \tau_1)h(t_2 - \tau_2)x(\tau_1)x(\tau_2)\mathrm{d}\tau_1\tau_2\right|h\right]$$

$$= \int_{\mathbb{R}^2} h(t_1 - \tau_1)h(t_2 - \tau_2)\underbrace{\mathbb{E}\left[x(\tau_1)x(\tau_2)\right]}_{=\sigma_x^2\delta(\tau_1 - \tau_2)}\mathrm{d}\tau_1\tau_2$$

$$= \sigma_x^2 \int_{\mathbb{R}} h(t_1 - \tau)h(t_2 - \tau)\mathrm{d}\tau.$$

Denoting $s = t_1 - \tau$ and $t = t_2 - t_1$ above, reveals that the kernel of $f|h$ is stationary and given by the convolution between the filter $h(t)$ and its mirrored version with respect to $t = 0$

$$\mathcal{K}_{f|h}(t_1, t_2) = \mathcal{K}_{f|h}(t) = \int_{\mathbb{R}} h(s)h(s + t)\mathrm{d}s = h(t) * h(-t). \tag{3}$$

## 2 Derivation of $\mathcal{K}_{f|\mathbf{h}}$ and $\mathcal{F}(\mathcal{K}_{f|\mathbf{h}(t)})$ — given $\mathbf{h} = h(\mathbf{t}) \in \mathbb{R}^{N_h}$

The kernel approximation $\mathcal{K}_{f|\mathbf{h}}(t_1, t_2)$ is given by:

$$\mathcal{K}_{f|\mathbf{h}}(t_1, t_2) = \mathbb{E}[\mathcal{K}_{f|h}(t_1, t_2)|\mathbf{h}] = \mathbb{E}\left[\left.\int_{\mathbb{R}} h(t_1 - \tau)h(t_2 - \tau)\mathrm{d}\tau\right|\mathbf{h}\right] =$$

$$= \int_{\mathbb{R}} \mathbb{E}\left[h(t_1 - \tau)h(t_2 - \tau)|\mathbf{h}\right]\mathrm{d}\tau$$

$$= \int_{\mathbb{R}} \mathrm{cov}(h(t_1 - \tau), h(t_2 - \tau)|\mathbf{h}) + \mathbb{E}\left[h(t_1 - \tau)|\mathbf{h}\right]\mathbb{E}\left[h(t_2 - \tau)|\mathbf{h}\right]\mathrm{d}\tau$$

$$= \int_{\mathbb{R}} K_{\mathrm{DSE}}(t_1 - \tau, t_2 - \tau) - K_{\mathrm{DSE}}(t_1 - \tau, \mathbf{t})\mathbf{K}^{-1}K_{\mathrm{DSE}}(\mathbf{t}, t_2 - \tau) + K_{\mathrm{DSE}}(t_1 - \tau, \mathbf{t})\mathbf{K}^{-1}\mathbf{h}\mathbf{h}^T\mathbf{K}^{-1}K_{\mathrm{DSE}}(\mathbf{t}, t_2 - \tau)\mathrm{d}\tau$$

$$= \int_{\mathbb{R}} K_{\mathrm{DSE}}(t_1 - \tau, t_2 - \tau)\mathrm{d}\tau + \int_{\mathbb{R}} K_{\mathrm{DSE}}(t_1 - \tau, \mathbf{t})\left(\mathbf{K}^{-1}\mathbf{h}\mathbf{h}^T\mathbf{K}^{-1} - \mathbf{K}^{-1}\right)K_{\mathrm{DSE}}(\mathbf{t}, t_2 - \tau)\mathrm{d}\tau$$

$$= \int_{\mathbb{R}} K_{\mathrm{DSE}}(t_1 - \tau, t_2 - \tau)\mathrm{d}\tau + \sum_{r,s=1}^{N_h} M_{r,s}\int_{\mathbb{R}} K_{\mathrm{DSE}}(t_1 - \tau, \mathbf{t}_r)K_{\mathrm{DSE}}(\mathbf{t}_s, t_2 - \tau)\mathrm{d}\tau$$

where $M_{r,s}$ is the $(r,s)^{\text{th}}$ entry of the matrix $(\mathbf{K}^{-1}\mathbf{h}\mathbf{h}^T\mathbf{K}^{-1} - \mathbf{K}^{-1})$, $\mathbf{K} = K_{\text{DSE}}(\mathbf{t}, \mathbf{t})$. The arguments of the integrals above are exponential functions of quadratic polynomials of the integration variable $\tau$; therefore, the kernel approximation admits a closed-form representation (see eq. (1)) and is stationary. Setting $t = t_1 - t_2$, we have:

$$\mathcal{K}_{f|\mathbf{h}}(t) = \frac{\sigma_g^2\sqrt{\pi}}{\sqrt{2\alpha}} e^{\left(-\frac{2\gamma+\alpha}{2}t^2\right)} + \frac{\sigma_g^4\sqrt{\pi}}{\sqrt{2(\alpha+\gamma)}} \sum_{r,s=1}^{N_{\mathbf{t}}} M_{r,s} e^{\left(-\frac{\alpha(\alpha+2\gamma)}{\alpha+\gamma}(t_r+t_s)^2 - \frac{\gamma+\alpha}{2}\left(t-\frac{\gamma}{\gamma+\alpha}(t_s-t_r)\right)^2\right)}.$$

The approximated kernel takes a parametric form — as is the case for the posterior mean of a Gaussian process — comprising a sum of **non-centred** squared exponential functions, where the centres and the weights are given by the locations $\mathbf{t}$, the vector $\mathbf{h}$ and the kernel $K_{\text{DSE}}$. The power spectral density can now also be computed by applying the Fourier transform to the above expression; this results in a mixture of Gaussians functional form, since the Fourier transform of a mixture of Gaussians is a mixture of Gaussians. This reveals that the DSE-GPCM as a spectral mixture (SM) [2] with an infinite number of components.

The power spectral density of the process $f|\mathbf{h}$, given the Fourier transform of the the kernel $\mathcal{K}_{\mathbf{h}}$, is a mixture of complex exponentials (due to the linearity of the Fourier transform and the footnote in the previous page), and can be computed in closed form too:

$$\mathcal{F}(\mathcal{K}_{f|\mathbf{h}(t)})(\omega) = \int_{\mathbb{R}} \mathcal{K}_{\mathbf{h}} e^{-2\pi j\omega t}\mathrm{d}t = \frac{\sigma_g^2\pi}{\sqrt{\alpha(2\gamma+\alpha)}} \exp\left(-\frac{\pi^2\omega^2}{\gamma+\alpha/2}\right)$$

$$+ \frac{\sigma_g^4\pi}{\alpha+\gamma} \sum_{r,s=1}^{N_g} M_{r,s} \exp\left(-\frac{\pi^2\omega^2}{(\gamma+\alpha)/2} - 2\pi j\omega\frac{\gamma}{\gamma+\alpha}(t_r-t_s) - \frac{\alpha(\alpha+2\gamma)}{\alpha+\gamma}(t_r+t_s)^2\right). \tag{4}$$

Observe that, in line with the symmetry of the spectrum of real-valued signals, the expression above is symmetric due to the term $t_r - t_s, r, s = 1..N_{\mathbf{h}}$.

# 3   Covariance of inducing variables

Since $u_h$ are chosen to be pseudo observations of the filter function $h$, they follow the same distribution, thus

$$\mathcal{K}_{u_h}(t, t') = \mathcal{K}_{h,u_h}(t, t') = \mathcal{K}_h(t, t') = K_{\text{DSE}}(t, t') = \sigma_h^2 \exp\left(-\gamma(t-t')^2 - \alpha t^2 - \alpha(t')^2\right). \tag{5}$$

For the white-noise process, however, we considered interdomain inducing variables given by $u_x = \sigma \int_{\mathbb{R}} x(\tau) \exp\left(\frac{-(\tau-t_x)^2}{2l^2}\right)\mathrm{d}\tau$, this yields

$$\mathcal{K}_{u_x}(t_x, t_x') = \mathbb{E}(u_x u_x') = \sqrt{\pi l^2}\sigma^2\sigma_x^2 \exp\left(\frac{-1}{4l^2}(t_x - t_x')^2\right) \tag{6}$$

$$\mathcal{K}_{u_xx}(t_x, t) = \mathbb{E}(u_x x(t)) = \sigma\sigma_x^2 \exp\left(\frac{-1}{2l^2}(t_i - t_x)^2\right). \tag{7}$$

# 4   Explicit form of the variational lower bound $\mathcal{F}$

The variational lowerbound is given by

$$\log p(\mathbf{y}) \geq \int q(x, h, u_x, u_h) \log \frac{p(\mathbf{y}|x,h)p(u_x)p(u_h)}{q(u_x)q(u_h)}\mathrm{d}x\mathrm{d}h\mathrm{d}u_x\mathrm{d}u_h \tag{8}$$

$$= \underbrace{\int q(x, h, u_x, u_h)\log p(\mathbf{y}|x,h)\mathrm{d}x\mathrm{d}u\mathrm{d}u_x\mathrm{d}u_h}_{\text{T1}} + \underbrace{\int \log \frac{p(u_x)p(u_h)}{q(u_x)q(u_h)}q(u_x)q(u_h)\mathrm{d}u_x\mathrm{d}u_h}_{\text{T2}} \tag{9}$$

where T1 is theaverage log-likelihood under $q(\cdot)$, and T2 is the negative KL divergence between $q(u_x, u_h)$ and $p(u_x, u_h)$. We next assume the parametric form for the variational distribution given by $q(u_x, u_h) = \mathcal{N}(u_x; \mu_x, C_x)\mathcal{N}(u_h; \mu_h, C_h)$. Also, we will denote the inducing locations of $h$ and $x$ by $\mathbf{t}_h = [t_{h,1}, \dots, t_{h,N_{\mathbf{h}}}]$ and $\mathbf{t}_x = [t_{x,1}, \dots, t_{x,N_{\mathbf{x}}}]$ respectively, and by $\{t_i\}_{i=1:N}$ the location of the data (i.e., $y_i = y(t_i)$).

## 4.1 Term T1 in (9)

Recall that $p(\mathbf{y}|x,h) = \prod_{i=1}^{N} p(y_i|x,h)$ to give

$$\int q(x,h,u_x,u_h) \log p(\mathbf{y}|x,h) \mathrm{d}x\mathrm{d}u\mathrm{d}u_x\mathrm{d}u_h = \int q(x,h,u_x,u_h) \sum_{i=1}^{N} \log p(y_i|x,h)\mathrm{d}x\mathrm{d}h\mathrm{d}u_x\mathrm{d}u_h$$

$$= \int q(x,h,u_x,u_h) \sum_{i=1}^{N} \log \left( \frac{1}{\sqrt{2\pi\sigma_y^2}} \exp \frac{\left(-\left(y_i - \int h(t_i - \tau)x(\tau)\mathrm{d}\tau\right)^2\right)}{2\sigma_y^2} \right) \mathrm{d}x\mathrm{d}h\mathrm{d}u_x\mathrm{d}u_h$$

$$= \sum_{i=1}^{N} \int q(x,h,u_x,u_h) \left( \frac{-1}{2}\log 2\pi\sigma_y^2 + \frac{\left(-\left(y_i^2 - 2y_i\int h(t_i-\tau)x(\tau)\mathrm{d}\tau + (\int h(t_i-\tau)x(\tau)\mathrm{d}\tau)^2\right)\right)}{2\sigma_y^2} \right)\mathrm{d}x\mathrm{d}u\mathrm{d}u_x\mathrm{d}u_h$$

$$= \frac{-N}{2}\log 2\pi\sigma_y^2 + \frac{-1}{2\sigma_y^2}\sum_{i=1}^{N} \int q(x,h,u_x,u_h) \left( y_i^2 - 2y_i\int h(t_i-\tau)x(\tau)\mathrm{d}\tau + \left(\int h(t_i-\tau)x(\tau)\mathrm{d}\tau\right)^2 \right)\mathrm{d}x\mathrm{d}u\mathrm{d}u_x\mathrm{d}u_h$$

$$= \frac{-N}{2}\log 2\pi\sigma_y^2 + \frac{-1}{2\sigma_y^2}\sum_{i=1}^{N} y_i^2 \qquad \textcolor{red}{\text{(constant term)}}$$

$$+ \frac{1}{2\sigma_y^2}\sum_{i=1}^{N} 2y_i \int h(t_i-\tau)x(\tau)q(x,h,u_x,u_h)\mathrm{d}x\mathrm{d}h\mathrm{d}u_x\mathrm{d}u_h\mathrm{d}\tau \qquad \textcolor{red}{\text{(linear term)}} \tag{10}$$

$$+ \frac{-1}{2\sigma_y^2}\sum_{i=1}^{N} \int h_{t_i-\tau_1}x_{\tau_1}h_{t_i-\tau_2}x_{\tau_2}q(x,h,u_x,u_h)\mathrm{d}x\mathrm{d}h\mathrm{d}u_x\mathrm{d}u_h\mathrm{d}\tau_1\mathrm{d}\tau_2 \qquad \textcolor{red}{\text{(quadratic term)}} \tag{11}$$

## 4.2 The integral in (10) — <span style="color:red">(linear term, LT)</span>.

$$\int h(t_i-\tau)x(\tau)q(x,h,u_x,u_h)\mathrm{d}x\mathrm{d}h\mathrm{d}u_x\mathrm{d}u_h\mathrm{d}\tau = \int h(t_i-\tau)p(h|u_h)\mathrm{d}hq(u_h)\mathrm{d}u_h x(\tau)p(x|u_x)\mathrm{d}xq(u_x)\mathrm{d}u_x\mathrm{d}\tau$$

$$= \int u_h^T K_h^{-1}(\mathbf{t}_h,\mathbf{t}_h)K_h(\mathbf{t}_h,t_i-\tau)q(u_h)\mathrm{d}u_h K_{u_x x}(\tau,\mathbf{t}_x)K_{u_x}^{-1}(\mathbf{t}_x,\mathbf{t}_x)u_x q(u_x)\mathrm{d}u_x\mathrm{d}\tau$$

$$= \mu_h^T K_h^{-1}(\mathbf{t}_h,\mathbf{t}_h)\underbrace{\left(\int K_h(\mathbf{t}_h,t_i-\tau)K_{u_x x}(\tau,\mathbf{t}_x)\mathrm{d}\tau\right)}_{I^{\text{linear}}} K_{u_x}^{-1}(\mathbf{t}_x,\mathbf{t}_x)\mu_x$$

where $(r,s)-$element of $I^{\text{linear}}$ is given by

$$I_{r,s}^{\text{linear}} = \int K_h(t_{h,r},t_i-\tau)K_{u_x x}(\tau,t_{x,s})\mathrm{d}\tau$$

$$= \sigma_h^2\sigma_x^2\sigma_u \int \exp\left(-\gamma(t_{h,r}-t_i+\tau)^2 - \alpha(t_{h,r}^2 + (t_i-\tau)^2) - \frac{1}{2l^2}(\tau-t_{x,s})^2\right)\mathrm{d}\tau$$

$$= \sigma_h^2\sigma_x^2\sigma_u \int \exp\left(-L\tau^2 + 2\left(\alpha t_i + \tfrac{1}{2l^2}t_{x,s} + \gamma(t_i-t_{h,r})\right)\tau - \tfrac{1}{2l^2}t_{x,s}^2 - \alpha(t_i^2 + t_{h,r}^2) - \gamma(t_i-t_{h,r})^2\right)\mathrm{d}\tau$$

where $L = \alpha + \gamma + \frac{1}{2l^2}$. Applying the identity in eq. (1) we obtain:

$$I_{r,s}^{\text{linear}} = \frac{\sigma_h^2\sigma_x^2\sigma_u\sqrt{\pi}}{\sqrt{L}}\exp\left(-\tfrac{1}{2l^2}t_{x,s}^2 - \alpha(t_i^2 + t_{h,r}^2) - \gamma(t_i-t_{h,r})^2 + \frac{\left(\alpha t_i + \tfrac{1}{2l^2}t_{x,s} + \gamma(t_i-t_{h,r})\right)^2}{L}\right). \tag{12}$$

## 4.3 The integral in (11) — (quadratic term, QT).

Let us first denote (see the covariance of the inducing variables in sec. 3)

$$\mathbf{M}^h = \mathbf{K}_{u_h}^{-1}(C_h + \mu_h\mu_h^T)\mathbf{K}_{u_h}^{-1} - \mathbf{K}_{u_h}^{-1}, \text{ where } \mathbf{K}_{u_h} = \mathcal{K}_{u_h}(\mathbf{t}_h, \mathbf{t}_h) \tag{13}$$

$$\mathbf{M}^x = \mathbf{K}_{u_x}^{-1}(C_x + \mu_x\mu_x^T)\mathbf{K}_{u_x}^{-1} - \mathbf{K}_{u_x}^{-1}, \text{ where } \mathbf{K}_{u_x} = \mathcal{K}_{u_x}(\mathbf{t}_x, \mathbf{t}_x). \tag{14}$$

$$\int h_{t_i-\tau_1}x_{\tau_1}h_{t_i-\tau_2}x_{\tau_2}q(x,h,u_x,u_h)\mathrm{d}x\mathrm{d}h\mathrm{d}u_x\mathrm{d}u_h\mathrm{d}\tau_1\mathrm{d}\tau_2$$

$$= \int h_{t_i-\tau_1}h_{t_i-\tau_2}p(h|u_h)q(u_h)\mathrm{d}h\mathrm{d}u_h x_{\tau_1}x_{\tau_2}p(x|u_x)q(u_x)\mathrm{d}x\mathrm{d}u_x\mathrm{d}\tau_1\mathrm{d}\tau_2$$

$$= \int \left(K_h(t_i-\tau_1,t_i-\tau_2) + K_h(t_i-\tau_1,\mathbf{t}_h)\mathbf{M}^h K_h(\mathbf{t}_h,t_i-\tau_2)\right)\left(K_x(\tau_1,\tau_2) + K_{xu_x}(\tau_1,\mathbf{t}_x)\mathbf{M}^x K_{u_xx}(\mathbf{t}_x,\tau_2)\right)\mathrm{d}\tau_1\mathrm{d}\tau_2$$

$$= \int K_h(t_i-\tau_1,t_i-\tau_2)K_x(\tau_1,\tau_2)\mathrm{d}\tau_1\mathrm{d}\tau_2 \qquad \text{(QT1)}$$

$$+ \int K_h(t_i-\tau_1,t_i-\tau_2)K_{xu_x}(\tau_1,\mathbf{t}_x)\mathbf{M}^x K_{u_xx}(\mathbf{t}_x,\tau_2)\mathrm{d}\tau_1\mathrm{d}\tau_2 \qquad \text{(QT2)}$$

$$+ \int K_h(t_i-\tau_1,\mathbf{t}_h)\mathbf{M}^h K_h(\mathbf{t}_h,t_i-\tau_2)K_x(\tau_1,\tau_2)\mathrm{d}\tau_1\mathrm{d}\tau_2 \quad \text{(QT3)}$$

$$+ \int K_h(t_i-\tau_1,\mathbf{t}_h)\mathbf{M}^h K_h(\mathbf{t}_h,t_i-\tau_2)K_{xu_x}(\tau_1,\mathbf{t}_x)\mathbf{M}^x K_{u_xx}(\mathbf{t}_x,\tau_2)\mathrm{d}\tau_1\mathrm{d}\tau_2 \qquad \text{(QT4)}$$

by replacing the kernels in eqs. (5), (6), (7), recalling that $x(t)$ is white Gaussian noise, and using the identites (1) and (2), we have:

$$\text{(QT1)} = \sigma_x^2 \int K_h(t_i-\tau,t_i-\tau)\mathrm{d}\tau$$

$$= \sigma_h^2\sigma_x^2 \int \exp\left(-\gamma_h\cancel{(t_i-\tau-t_i+\tau)^2} - \alpha_h((t_i-\tau)^2 + (t_i-\tau)^2)\right)\mathrm{d}\tau$$

$$= \sigma_h^2\sigma_x^2 \int \exp\left(-2\alpha_h(t_i-\tau)^2\right)\mathrm{d}\tau$$

$$= \sigma_h^2\sigma_x^2\sqrt{\frac{\pi}{2\alpha_h}}$$

$$\text{(QT2)} = \int K_h(t_i-\tau_1,t_i-\tau_2)K_{xu_x}(\tau_1,\mathbf{t}_x)\mathbf{M}^x K_{u_xx}(\mathbf{t}_x,\tau_2)\mathrm{d}\tau_1\mathrm{d}\tau_2$$

$$= \sum_{r,s}\mathbf{M}_{r,s}^x \int K_h(t_i-\tau_1,t_i-\tau_2)K_{xu_x}(\tau_1,t_{x,r})K_{u_xx}(t_{x,s},\tau_2)\mathrm{d}\tau_1\mathrm{d}\tau_2$$

$$= \sum_{r,s}\mathbf{M}_{r,s}^x\sigma^2\sigma_u^2\sigma_x^4 \underbrace{\int \exp\left(-\gamma(\tau_1-\tau_2)^2 - \alpha((t_i-\tau_1)^2 + (t_i-\tau_2)^2) - \tfrac{1}{2l^2}((\tau_1-t_{x,r})^2 + (\tau_2-t_{x,s})^2)\right)\mathrm{d}\tau_1\mathrm{d}\tau_2}_{I_{r,s,t_i}^{\text{QT2}}},$$

where

$$I_{r,s,t_i}^{\text{QT2}} = \int \exp\left(-L\tau_1^2 - L\tau_2^2 + \underbrace{2\gamma}_{b_{12}}\tau_1\tau_2 + \underbrace{\left(2\tfrac{1}{2l^2}t_{x,r} + 2\alpha t_i\right)}_{b_1}\tau_1 + \underbrace{\left(2\tfrac{1}{2l^2}t_{x,s} + 2\alpha t_i\right)}_{b_2}\tau_2 \underbrace{-2\alpha t_i^2 - \tfrac{1}{2l^2}(t_{x,r}^2 + t_{x,s}^2)}_{c_1}\right)\mathrm{d}\tau_1\mathrm{d}\tau_2$$

$$= \frac{2\pi}{\sqrt{4L^2 - b_{12}^2}}\exp\left(c_1 + \frac{L(b_1^2 + b_2^2) + b_1 b_{12} b_2}{4L^2 - b_{12}^2}\right)$$

$$(\text{QT3}) = \int K_h(t_i - \tau_1, \mathbf{t}_h) \mathbf{M}^h K_h(\mathbf{t}_h, t_i - \tau_2) K_x(\tau_1, \tau_2) \mathrm{d}\tau_1 \mathrm{d}\tau_2$$

$$= \sigma^4 \sigma_x^2 \frac{\sqrt{\pi}}{\sqrt{2\gamma + 2\alpha}} \mathrm{Tr} \left\{ \mathbf{M}^h \mathbf{I}^{\text{QT3}} \right\}, \quad \text{where } I_{r,s}^{\text{QT3}} = \exp \left( -(t_{h,s}^2 + t_{h,r}^2)(\alpha + \gamma) + \frac{(\gamma(t_{h,r} + t_{h,s}))^2}{2(\gamma + \alpha)} \right)$$

$$(\text{QT4}) = \int K_h(t_i - \tau_1, \mathbf{t}_h) \mathbf{M}^h K_h(\mathbf{t}_h, t_i - \tau_2) K_{x u_x}(\tau_1, \mathbf{t}_x) \mathbf{M}^x K_{u_x x}(\mathbf{t}_x, \tau_2) \mathrm{d}\tau_1 \mathrm{d}\tau_2$$

$$= \int \sum_{p,q,r,s} K_h(t_i - \tau_1, t_{h,p}) \mathbf{M}_{p,q}^h K_h(t_{h,q}, t_i - \tau_2) K_{x u_x}(\tau_1, t_{x,r}) \mathbf{M}_{r,s}^x K_{u_x x}(t_{x,s}, \tau_2) \mathrm{d}\tau_1 \mathrm{d}\tau_2$$

$$= \sigma^4 \sigma_x^4 \sigma_u^2 \sum_{p,q,r,s} \mathbf{M}_{p,q}^h \mathbf{M}_{r,s}^x \int \exp \left( -\gamma((t_i - \tau_1 - t_{h,p})^2 + (t_{h,q} - t_i + \tau_2)^2) \right)$$

$$\cdot \exp \left( -\alpha((t_i - \tau_1)^2 + t_{h,p}^2 + (t_i - \tau_2)^2 + t_{h,q}^2) \right)$$

$$\underbrace{\cdot \exp \left( -\frac{1}{2l^2}((\tau_1 - t_{x,r})^2 + (\tau_2 - t_{x,s})^2) \right) \mathrm{d}\tau_1 \mathrm{d}\tau_2}_{I_{p,q,r,s,t_i}^{\text{QT4}}}$$

where

$$I_{p,q,r,s,t_i}^{\text{QT4}} = \int \exp \Big( -L\tau_1^2 - L\tau_2^2 + \underbrace{0}_{b_{12}} \tau_1 \tau_2 + \underbrace{(2\alpha t_i + 2\tfrac{1}{2l^2} t_{x,r} + \gamma 2(t_i - t_{h,p}))}_{b_1} \tau_1 + \underbrace{(2\alpha t_i + 2\tfrac{1}{2l^2} t_{x,s} + \gamma 2(t_i - t_{h,q}))}_{b_2} \tau_2$$

$$\underbrace{-\frac{1}{2l^2}(t_{x,s}^2 + t_{x,r}^2) - \gamma((t_i - t_{h,p})^2 + (t_i - t_{h,q})^2) - \alpha(2t_i^2 + t_{h,p}^2 + t_{h,q}^2)}_{c_1} \Big) \mathrm{d}\tau_1 \mathrm{d}\tau_2$$

$$= \frac{\pi}{L} \exp \left( c_1 + \frac{L b_1^2 + b_2^2 L}{4L^2} \right) = \frac{\pi}{L} \exp \left( c_1 + \frac{b_1^2 + b_2^2}{4L} \right)$$

## 4.4 Term T2 in $(9)$ — (-KLs)

$$\int \log \frac{p(u_h)}{q(u_h)} q(u_h) \mathrm{d}u_h = \int \log \left( \frac{|\mathbf{K}_h|^{-1/2}(2\pi)^{-N_h/2} \exp(-1/2 u_h^T \mathbf{K}_h^{-1} u_h)}{|C_h|^{-1/2}(2\pi)^{-N_h/2} \exp(-1/2(u_h - \mu_h)^T C_h^{-1}(u_h - \mu_h))} \right) q(u_h) \mathrm{d}u_h$$

$$= \log \frac{\sqrt{|C_h|}}{\sqrt{|\mathbf{K}_h|}} + \int \left( \frac{-1}{2} u_h^T \mathbf{K}_h^{-1} u_h + \frac{1}{2}(u_h - \mu_h)^T C_h^{-1}(u_h - \mu_h) \right) q(u_h) \mathrm{d}u_h$$

$$= \log \frac{\sqrt{|C_h|}}{\sqrt{|\mathbf{K}_{u_h}|}} - \frac{1}{2} \left( \mu_h^T \mathbf{K}_{u_h}^{-1} \mu_h + \mathrm{Tr} \left\{ \mathbf{K}_{u_h}^{-1} C_h \right\} \right) + \frac{1}{2} N_{\mathbf{h}} \qquad \text{(KLT1)}$$

and similarly

$$\int \log \frac{p(u_x)}{q(u_x)} q(u_x) \mathrm{d}u_x = \log \frac{\sqrt{|C_x|}}{\sqrt{|\mathbf{K}_{u_x}|}} - \frac{1}{2} \left( \mu_x^T \mathbf{K}_{u_x}^{-1} \mu_x + \mathrm{Tr} \left\{ \mathbf{K}_{u_x}^{-1} C_x \right\} \right) + \frac{1}{2} N_{\mathbf{x}} \qquad \text{(KLT2)}$$

## 4.5   Summary:

The explicit form of the variational lowerbound is then given by the sum of the follwing terms

$$\text{(Constant term)} \qquad \text{CT} = \frac{-N}{2}\log 2\pi\sigma_y^2 + \frac{-1}{2\sigma_y^2}\sum_{i=1}^{N} y_i^2$$

$$\text{(Linear term)} \qquad \text{LT} = \frac{1}{2\sigma_y^2}2\mu_h^T K_h^{-1}(\mathbf{t}_h,\mathbf{t}_h)\left(\sum_{i=1}^{N} y_i \mathbf{I}_{t_i}^{\text{linear}}\right)K_{u_x}^{-1}(\mathbf{t}_x,\mathbf{t}_x)\mu_x$$

$$\text{(Quadratic terms)} \qquad \text{QT1} = \frac{-N\sigma_h^2\sigma_x^2}{2\sigma_y^2}\sqrt{\frac{\pi}{2\alpha}}$$

$$\text{QT2} = \frac{-\sigma^2\sigma_u^2\sigma_x^4}{2\sigma_y^2}\text{Tr}\left\{\left(\sum_{i=1}^{N}\mathbf{I}_{t_i}^{\text{QT2}}\right)\mathbf{M}^x\right\}$$

$$\text{QT3} = \frac{-N\sigma^4\sigma_x^2\sqrt{\pi/2}}{2\sigma_y^2\sqrt{\gamma+\alpha}}\text{Tr}\left\{\mathbf{M}^h\mathbf{I}^{\text{QT3}}\right\}$$

$$\text{QT4} = \frac{-\sigma^4\sigma_x^4\sigma_u^2}{2\sigma_y^2}\sum_{i=1}^{N}\sum_{p,q,r,s}\mathbf{M}_{p,q}^h\mathbf{M}_{r,s}^x I_{p,q,r,s,t_i}^{\text{QT4}}$$

$$\text{(-KL terms)} \qquad \text{KLT1} = \log\frac{\sqrt{|C_h|}}{\sqrt{|\mathbf{K}_{u_h}|}} - \frac{1}{2}\left(\mu_h^T\mathbf{K}_{u_h}^{-1}\mu_h + \text{Tr}\left\{\mathbf{K}_{u_h}^{-1}C_h\right\}\right) + \frac{1}{2}N_{\mathbf{h}}$$

$$\text{KLT2} = \log\frac{\sqrt{|C_x|}}{\sqrt{|\mathbf{K}_{u_x}|}} - \frac{1}{2}\left(\mu_x^T\mathbf{K}_{u_x}^{-1}\mu_x + \text{Tr}\left\{\mathbf{K}_{u_x}^{-1}C_x\right\}\right) + \frac{1}{2}N_{\mathbf{x}}$$

Note that all terms above correspond to inverse matrices and linear combinations of exponential functions, therefore, all the derivaties can be calculated analytically, meaning that the bound can be maximised using conjugate gradients.