[Reviews · NeurIPS 2015]

Submitted by Assigned_Reviewer_1

This paper proposes a generative model for time series data that models data using a Gaussian process with "nonparametric kernels".

In the standard approach to GP regression,kernels are hand-designed to be of a particular parametric form, with hyperparameters set by maximum likelihood learning.

The proposed method defines a nonparametric prior over the kernel that can be learned using a variational approach, and does not explicitly limit the form of the kernel.

Experiments on both synthetic and real-world data are presented.

The work is technically sound as far as I can tell.

The prior over nonparametric kernels is induced by assuming the data is generated by an underlying function (f) that is a convolution between a Gaussian process (h) and a white noise process (x).

The Gaussian process h is assumed to have a standard parametric covariance function, but samples from h determines the covariance of f.

The manuscript is generally well-written and well-organized, but assumes substantial background on variational approximation and signal processing.

There are minor typos that should be fixed.

For example,

line 318: Fig. 4.1 should be figure 2 line 387: Figure 4 captions should not include Top or Bottom

The model and variational approximation proposed in the work is novel and interesting.

I'm not aware of similar approaches to modeling time series in the GP literature.

The paper does a good job of positioning itself with respect to alternative approaches to kernel design, but could benefit from some discussion of prior GP time series models.

I'm interested in why there are no attempts to evaluate the ability of the model to perform forecasting on real-world data.

How would the proposed model compare against a simple GP regression model using a RBF kernel?

I feel the significance of the work could be improved if more compelling practical applications are demonstrated.

Alternatively, it would be useful to say more about why accurately recovering spectral densities from time-series data is an important problem and how the current method compares to the state of the art for this task.

Summary: This paper proposes a generative model for time series data that leads to a Gaussian process with nonparametric kernels. A variational approach is taken to learn the model, experiments on both synthetic and real-world data are presented.

Submitted by Assigned_Reviewer_2

The paper is concerned with the important problem of analyzing impulse responses of LTI systems with GPs. I feel that the paper is nice, but it does not really project the methods to existing approaches which would be an important thing to do. This is a classical problem anyhow. Some comments:

- In Section 2 the authors talk about "convergence" without any reference to the mode of convergence or references to literature. Certainly the convergence properties of the integral (1) are known, e.g., in suitable Schwartz spaces because Wiener alone wrote tens of books on harmonic analysis.

- Based on the text I would guess that the authors wish to talk about mean square convergence, which is related to Ito's work (stochastic calculus) and where the integral (1) needs to be defined as an Ito integral f(t) = int h(t - tau) dW(tau) which is how it is classically done (see, e.g., the refs. below).

- The problem considered here is called "system identification" (cf. google), and it certainly is not a new problem. This problem is classically treated in the framework of "Kalman filtering", which is "GP regression for time series" and works in the Ito calculus / state-space formalism. Nice classic textbooks on the core subject are, e.g., the following:

"A. H. Jazwinski (1970). Stochastic processes and filtering theory." "P. Maybeck (1979). Stochastic models, estimation, and control. Volumes 1&2."

- The basic classical method is indeed to use a parametric model for the impulse response h(t) instead of a non-parametric model. Typically this impulse response is constructed in a linear state-space form which is then parametrized suitably. State-space methods such as Kalman filtering based approaches can then be used for parameter estimation directly although the resulting model can be treated with other types of methods as well. To clarify: the Kalman filter can be used to evaluate the marginal likelihood using the prediction error decomposition and conventional parameter inference can be implemented on top of it.

- It would be beneficial to compare at least to a very simple parametric model to the impulse response---or at least discuss them. You could, for example, use an all-pole type of transfer function model

H(s) = b0 / (a0 + a1 s + a2 s^2)

where the parameters a0,a1,a2,b0 are the free parameters. The inverse Laplace transform of this then implicitly defines the impulse response although it is easier to just convert this into state space form and use a Kalman filter type of method to estimate b0,a0,a1,a2. A more elaborate model would be a higher order rational function model.

- One point is that DSP books such as Oppenheim et al. does not treat the above kind of models, but you need to look into (oldish) control engineering and system identification literature to find them---although some newer literature might exist. Just to point out that using the classical state-space/Ito approach above, the non-uniform sampling is *not* a problem at all even though from Oppenheim et al it might seem to be.

- However, system identification problems are indeed classically parametric and it is a difference to here and the novelty of the article. Furthermore, the above approach for system identification has number of difficulties, for example, it is quite hard to enforce the transfer function to correspond to a stable system. Thus new methods are welcome. In any case, it is not an excuse to ignore the existing literature, you should at least discuss them.

- The approach itself is very simple which is nice. It also points out many connections to inducing point methods which is nice.

- The weakness in the experiment section is also the lack of comparison to classical approaches.

Summary: The approach is interesting and might even be novel. The paper does not discuss, cite, or compare to classical approaches to the same (system identification) problem.

Submitted by Assigned_Reviewer_3

This paper takes a novel approach time series modeling, using a non-parametric kernel within a GP framework. The theoretical development draws interesting connections with SM kernels, and develops interesting new variational inference approaches.

Quality: the theoretical development is good, but the presentation of the experiments can be strengthened. (See below.)

Clarity:

Lines 311-314: can your method handle non-regularly spaced time series? You only consider SE and SM. But SE is not a good choice for most time series. I would consider Matern-1/2 and Matern-3/2.

There is no Figure 4.1. I guess you mean Figure 2. Please add labels to each subfigure like (a) and (b) because your references to subfigures is inconsistent.

Line 318: the term inducing points here confused me. Why are you calling the missing observations inducing points?

Figure 3: what filter and process did you use? The same ones shown in Figure 2?

You imputed the missing middle section, and compared to the SM kernel estimate. But why not show the real data? Isn't that what you're trying to recover? I guess the real data is just a noisy version of the green "Ground truth" line?

Section 4.2: was there some reason you couldn't put the linear trend in the mean function of the GP? And out of curiosity what would happen if you tried to fit something with a linear trend with your method?

Why are you reminding us about the Nyquist frequency? Where are the above 6/year found in Figure 4 (right).

You make a big point, I think, that your method gives posterior uncertainty intervals for the recovered kernel, i.e. there's no uncertainty intervals for SM in figure 4 (right). But you could have used a fully Bayesian approach (MCMC) with priors over the SM hyperparameters and thus learned a posterior over the SM kernel.

Figure 4 (left) shows that SM does a better job than your method. I don't think you should sweep this under the rug, nor do I think it invalidates your results. But you definitely should explain what you think is going on!
Summary: This paper's approach is novel and very interesting. The presentation could be made clearer, and the experiments definitely should be clarified.

Author Feedback
Author rebuttal: We thank the Reviewers for their constructive comments. We believe the model proposed is very powerful and theoretically deep. We agree with the reviewers that the exposition and experiments should be improved and will address this in the revision.

Q1. Experiments
The reviewers suggested improvements to the experiments and we agree. We will include extrapolation tests and comparisons to:
-GPs with Gauss-Markov and Matern kernels
-standard spectral estimation (incl. periodogram variants & Bayesian methods).

Reviewer_1
Q2: Forecasting on real-world data and RBF kernel.
A2: Our aim is to estimate auto-covariances and spectral densities; therefore, the real-data experiments focussed on inferring these quantities. It is, however, simple to handle forecasting (algorithmically identical to interpolation, Figs2&3). The RBF kernel makes short-range predictions reverting to the mean rapidly, whereas the new model can make long-range predictions.

Q3: Why accurate recovery of spectral densities (SD) is needed.
A3: SD estimation is key in source separation, denoising, compression, and imputation. Comparison to the state-of-the-art on real signals is challenging as the true spectrum is unknown, but the consistency tests in section 4.2 provide one opportunity. We will add comparisons.

Q4: Expand discussion of GP time-series models.
A4: Thanks. We will do this.

Reviewer_2

Q5: The paper does not discuss classical approaches.
A5: The goal of the paper is to extend the standard GP toolbox to specify covariance functions in a flexible non-parametric way and it was written for an audience familiar with GPs. We focussed on time-series modelling which connects to a number of important classical methods such as system-ID for estimating impulse response functions (control community) and the dual problem of spectrum estimation (signal processing community). We regret we did not fully layout these important connections, but will expand below and in the revision:

1. the model is a continuous-time generalisation of the moving average process (itself an important novel contribution) in which the window has infinite support and is non-parametric. The impulse response function is not constrained to be causal, unlike in many traditional approaches (it is more akin to a spatial impulse response, rather than temporal). A system sampled from the new model is guaranteed to be stable by construction (unlike e.g. Gauss-Markov models - see A7).

2. in the frequency domain the model defines a GP with a spectrum comprising an infinite mixture of Gaussians. Model fitting is therefore a form of non-parametric Bayesian spectrum estimation, but unlike many approaches to spectrum estimation, our method defines a valid probabilistic time-series model that allows imputation, extrapolation, etc.

3. importantly the framework is not limited to time-series, being applicable whenever a stationary covariance function is required. It generalises to higher dimensional inputs (e.g. space). We envisage applications out of reach of classical methods.

Q6: The paper should compare to parametric impulse response methods (Gauss-Markov processes, GM).
A6: We thank the reviewer for this suggestion and will add it. We are aware of these models (they are GPs after all, see Rasmussen&Williams app.B). In the authors' practical experience in discrete time systems, GM covariance functions perform very similarly to the spectral mixture ones (when number of poles equals number of spectral mixture components).

Q7: Stochastic integrals, convergence and stability.
A7: Eq. (1) is indeed a stochastic Ito integral. For clarity, we used a notation familiar to the GP community, but will add clarification. By convergence we mean E[f(t)^2]=K_DSE < infty, and observe that due to the Chebyshev inequality, this implies Pr(|f(t)| > L)= < K_DSE/L^2 - see eqs. (1-3). We will include this convergence property in the paper.

Reviewer_3

Q8: Irregular sampling.
A8: All experiments used observations randomly chosen from the data (i.e., irregular).

Q9: Line 318: inducing points.
A9: Inducing points are the same inducing variables (see Sec 3.1).

Q10: fig 3.
A10: The setup in Fig 2&3 is identical, with data from SE and SM kernels respectively.

Q11: Real data, Sec 4.2.
A11: The real data (observations) are a noisy version of the ground truth (the inferential goal), not shown in the imputation region to see the estimates clearly.

Q12: Removal of linear trend.
A12: Yes, it is simple to add the linear trend into the mean function and treat it in a fully Bayesian way - this would require a filter as long as the data.

Q13: Posterior over the SM kernel using MCMC.
A13: We found MCMC samplers struggle to mix for the SM variables, whereas the variational non-parametric method is faster and more flexible.

Q14: Why SM does better on Mauna Loa
A14: The SM identified sharper line spectra, the non-parametric method over-smoothed slightly.